# Highly efficient current-induced domain wall motion in a room temperature van der Waals magnet

Yicheng Guan[1,2], Yufeng Wu[1,2], Yan Zhang[1], Jae-Chun Jeon [1], Wenjie Zhang[1], Ke Xiao[1] & Stuart. S. P. Parkin [1] ✉

Two-dimensional van der Waals magnets are highly promising for next-generation spintronics. The ferromagnetic material $Fe_3GaTe_2$ is especially interesting due to its high Curie temperature. Here we demonstrate highly efficient current-induced domain wall motion in $Fe_3GaTe_2$ racetracks via spin-transfer torque that gives rise to the highest domain wall velocity yet reported for any van der Waals magnet. The spin polarization of the conduction electrons was measured via superconducting point-contact measurements revealing that 100% of their spin angular momentum is transferred to the domain walls. The very low threshold current density plus the very high mobility of the domain walls is attributed to the structural perfection of the two-dimensional magnet. We further demonstrate an electrically readable memristive racetrack device with more than four data bits, via precise domain wall positioning. Our work demonstrates that van der Waals magnets are compelling for emerging spintronic applications from room temperature to cryogenic temperatures.

Since the initial discovery of magnetic ordering in two-dimensional van der Waals (2D vdW) materials[1–7], a wide variety of 2D magnets have been discovered including ferromagnets[5,6,8,9], antiferromagnets[10–12] and spin glasses[13–15]. Recently, a 2D vdW magnet $Fe_3GaTe_2$ (FGaT) has been reported to be a ferromagnetic metal with a Curie temperature of almost 380 K, accompanied by a strong perpendicular magnetic anisotropy (PMA)[7]. Several studies have explored topological spin textures found in this material[16–18] but there have only been a limited number of studies on spin transport related phenomena and devices[19–21]. Racetrack memory[22–26] based on the current-induced motion of domain walls (CIDWM) along magnetic nanowires is one of the most interesting emerging spintronic devices and 2D vdW materials could have significant advantages, e.g., low power consumptions[27,28], for such devices.

Here, we present the first observation of the current-induced motion of domain walls (DWs) in FGaT racetracks. A very efficient spin-transfer torque (STT) driven motion of domain walls is found over a wide range of temperatures from room temperature to cryogenic temperatures. The domain wall velocity reaches values of 25 m s[1] below 50 K, which is the fastest DW velocity yet recorded in any vdW material. Furthermore, an ultra-low threshold current density to move a domain wall of just a few MA cm[−2] as well as a high DW mobility are observed. The threshold current density is one order of magnitude smaller, while the DW mobility is several times higher as compared to those in conventional ferromagnetic thin films and heterostructures. We find that the spin polarization of FGaT, as determined by superconducting point contact measurements at 2 K, is large and is comparable to that previously reported in 3 $d$ transition metal ferromagnets. An analysis based on a 1D-STT model reveals that in FGaT, the non-adiabatic term, $\beta$, is much larger than the Gilbert damping parameter, $\alpha$, indicating that the domain walls move even faster than would be anticipated from perfect spin angular-momentum transfer. The STT-driven DWM in FGaT is used to demonstrate an energy-efficient prototype racetrack memristor

[1]Max-Planck Institute for Microstructure Physics, Halle (Saale), Germany. [2]These authors contributed equally: Yicheng Guan, Yufeng Wu.
✉e-mail: stuart.parkin@mpi-halle.mpg.de

device operated at ultra-low current densities based on the high-precision positioning of DWs.

## Results

Flakes of FGaT ~25 nm thick are obtained by exfoliation from a bulk single crystal. Racetrack devices, with a typical length of 30 μm and a width of 5.8 μm, are fabricated from the flakes using conventional electron beam lithography methods. Electrical contacts are formed using a lift-off technique (see "Methods" for more details). The CIDWM is studied using Kerr microscopy. Exemplary Kerr images of a device and CIDWM at ambient temperature (~290 K) are shown in Fig. 1a, b. A succession of Kerr images of a single DW are shown in Fig. 1b. Between each image a series of 20 current pulses with a 5 ns pulse length and a pulse amplitude of 3.2 MA cm$^{-2}$ are injected. The DW moves in the electron flow direction, which reflects a majority-spin polarized current that has often been observed for STT-driven DWM in conventional ferromagnetic systems[29,30]. The DW velocity, $v$, increases with current density, $J$, above a threshold current density, which is as low as ~2.8 MA cm$^{-2}$. The maximum velocity observed of ~8.4 m s$^{-1}$ is limited by Joule heating but, nevertheless, exceeds the fastest DW velocity yet reported in any 2D vdW magnet at room temperature. Above this current density the device forms a multi-domain state and, at yet higher densities, the temperature of the device exceeds its Curie temperature ($T_C$), which is estimated to be ~345 K from Reflective Magnetic Circular Dichroism (RMCD) measurements (see Fig. 1c and Fig. S1). The DWs are shown to be Néel type from the longitudinal in-plane field dependence of the DW velocity for up/down and down/up DWs (See Supplementary Note 1 and Fig. S2).

As the temperature is decreased, the velocity of the CIDWM is significantly enhanced because the sample can endure larger current densities before the thermal nucleation of magnetic domains takes place. The DW velocity, $v$, plotted as a function of injected current density, $J$, at various temperatures from 290 K to 50 K, is shown in Fig. 1d. The fastest CIDWM velocity that is observed is ~25 m s$^{-1}$ at an injected current density of ~9 MA cm$^{-2}$ at 50 K. In order to accurately determine the threshold current density, as well as quantifying the STT, the following formula is used, $v = u \times \sqrt{(J - J_{th})^2}$, where $J_{th}$ is the threshold current density and $u$ is the STT-driven DW mobility[31,32]. As shown in Figs. S1 and 1c, both $J_{th}$ and $u$ first decrease and then increase as the temperature is reduced. The high values of both $J_{th}$ and $u$ at 290 K are attributed to the small saturation magnetization, as the temperature is close to $T_C$ due to current induced heating. Nevertheless, the threshold current density is one order of magnitude smaller than those observed in conventional ferromagnetic thin film systems which are prepared by thin film deposition methods such as magnetron sputtering[23,24,30,33]. It is also worth noting that the threshold current density in FGaT is several times smaller than the previously reported STT-driven DWM in its sister material, Fe$_3$GeTe$_2$. The temperature dependence of $J_{th}$ is also considerably weaker. The low $J_{th}$ and its weak temperature dependence can be attributed to the atomic level smoothness of the FGaT layers that limits the spatial variation of magnetic properties, such as uniaxial magnetic anisotropy, which is considered to be the extrinsic origin of the high threshold current densities in ultra-thin magnetic heterostructures[33]. As previously reported, the threshold current density in magnetic heterostructures, e.g., Co/Pt, drastically increases with decreasing temperature, thereby limiting their applications at cryogenic temperatures[33]. For device applications, a low $J_{th}$ is very important, which, therefore, makes 2D vdW magnets such as FGaT very appealing, especially for cryogenic applications.

We now consider the DW mobility, $u$, which is also highly relevant to technological applications. In the DW flow regime, the DW mobility

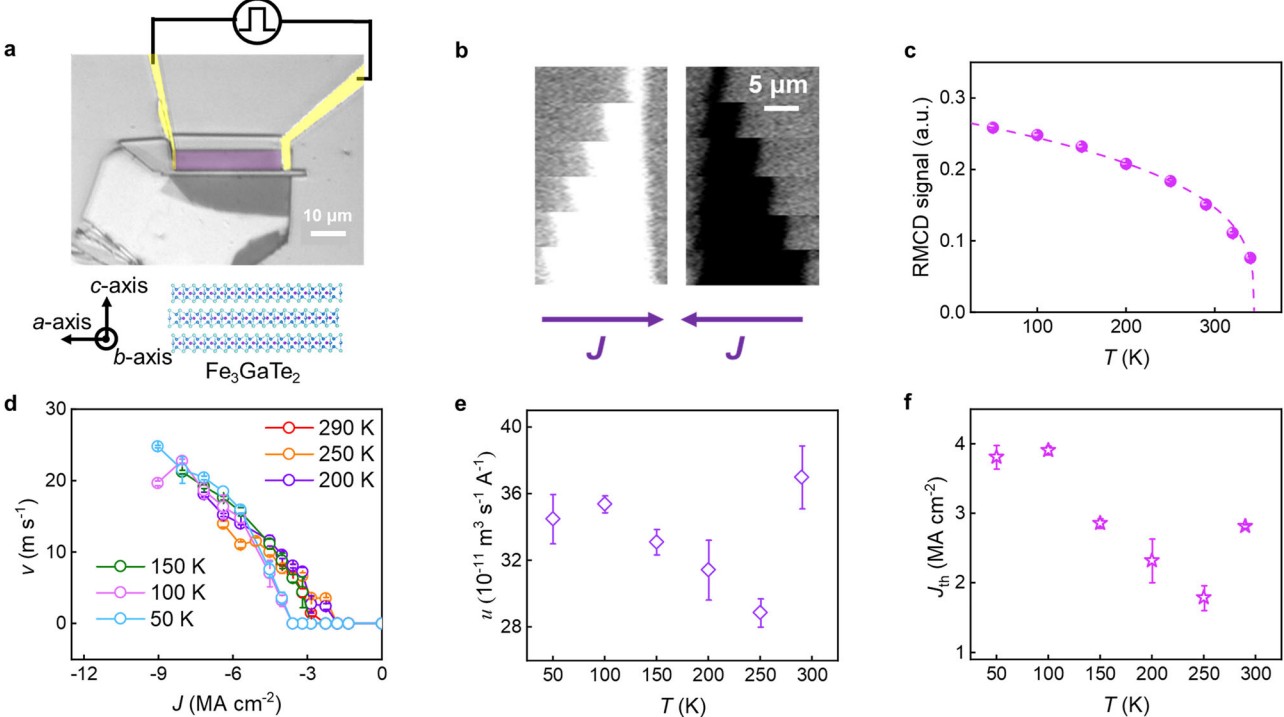

**Fig. 1 | Current-induced domain wall motion in Fe$_3$GaTe$_2$ nanoflakes. a** Upper: Kerr microscope image of the Racetrack device fabricated from a FGaT nanoflake. The violet area is the fabricated Racetrack device, isolated from the exfoliated flake via ion milling. Lower: Schematic illustration of the structure of FGT, showing the orientation of the crystal axes. **b** Exemplary images of the current induced motion of a single DW in the device shown in (**a**) at 290 K. The images show the position of the DW in response to bursts of 20 current pulses each with a magnitude of 3.2 MA cm$^{-2}$ and a length of 5 ns that are injected between successive images. **c** RMCD signal from the FGaT Racetrack versus temperature. The dotted line is a fit to the experimental data (solid circles) of the form $M(T) = M_S^0 \times (1 - \frac{T}{T_C})^\tau$. **d** DW velocity, $v$, versus injected current density, $J$, at several temperatures. Temperature dependence of (**e**) DW mobility, $u$, and (**f**) threshold current density, $J_{th}$. All error bars in (**c, d**) correspond to 1 SD.

**Table 1 | Domain wall mobilities for typical ferromagnetic systems**

|  | Permalloy Ref. [29] | Co/Ni Ref. [30] | Pt/Co Ref. [33] | Pt/SAF Ref. [33] | Pt/DL/SAF Ref. [33] | FGaT This Work |
|---|---|---|---|---|---|---|
| Measuring temperature (K) | 290 | 290 | 290 | 290 | 290 | 290 |
| DW mobility($10^{-11}\,A^{-1}\,m^3\,s^{-1}$) | 7.33 | 5.13 | 12 | 26 | 55 | 37 |
| Mechanism | Spin transfer torque | Spin transfer torque | Spin-orbit torque | Exchange-coupling torque | Exchange-coupling torque | Spin transfer torque |

Domain wall mobilities obtained in permalloy (Ref. [29]), Co/Ni (Ref. [30]), Pt/Co (Ref. [33]), Pt/SAF (Ref. [33]), Pt/DL/SAF (DL refer to dusting layer, Ref. [33]) and FGaT (this work) are listed together with their driving mechanism.

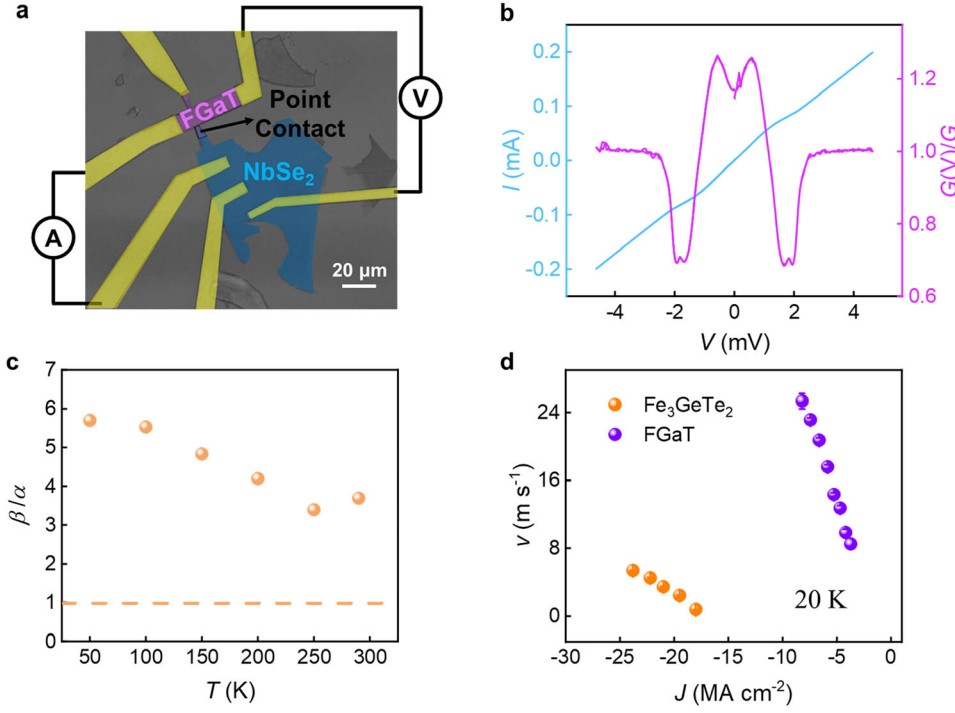

**Fig. 2 | Determination of spin polarization, $P$, and calculation of the ratio, $\beta/\alpha$. a** Microscopic image of a point contact formed between 2H-NbSe$_2$ and FGaT and schematic illustration of the measurement setup. **b** Current, $I$, versus applied voltage, $V$, at 2 K from which the differential conductance $G(V)$ is calculated. **c** Temperature dependence of the ratio of the non-adiabatic STT torque to the Gilbert damping ratio, $\beta/\alpha$, obtained from the 1D STT-driven DW model. The dashed line corresponds to $\beta/\alpha = 1$. **d** Comparison of CIDWM in Fe$_3$GeTe$_2$ (Ref. [26]) and FGaT at 20 K. The error bars in **d** correspond to 1 SD.

is given by $u \sim \frac{dv}{dJ}$. Compared to previous results of CIDWM in conventional ferromagnets, the DW mobility in FGaT is substantially higher, higher even than the highest value of $u$ previously reported in exchange coupling torque-driven DWM in synthetic antiferromagnetic (SAF) racetracks[29,30,33,34], (see Table 1). The $u$ at 290 K of ~ $38 \times 10^{-11}\,A^{-1}$ m$^3$ s$^{-1}$ would imply that the DW velocity could reach $100$ m s$^{-1}$ at a current density as low as 26 MA cm$^{-2}$ were it not for the comparatively low $T_C$ of FGaT. Thus, it will be very exciting to pursue vdW magnets with higher $T_C$, so that the CIDWM can be extended to higher current densities.

In the following we discuss the underlying mechanism for the highly efficient CIDWM in FGaT. A one-dimension model is often used to describe STT-driven DWM in which the form of the DW is assumed to be unchanged during its motion. Two torques contribute to the motion, an adiabatic and a non-adiabatic torque. When only the adiabatic torque is present, the spin angular momentum carried by the spin polarized current is transferred from the conduction electrons to the DW. The STT DW mobility is then given by $u = u_B P/e M_S$, where $u_B$, $P$, $e$, and $M_S$ are the Bohr magnetron, the spin polarization of the current, the electron charge and the saturation magnetization, respectively. This relation assumes the perfect

transfer of spin angular momentum from the current to the DW. By using the relationship, $M(T) = M_S^0 \times (1 - \frac{T}{T_C})^\tau$, where $M_S^0$ is the saturation magnetization at $T = 0$ and $\tau$ is the critical exponent, the $M_S$ value at each temperature can be deduced. From RMCD measurements on our flake, we obtain $T_C = $ ~34 K and $\tau$ ~0.3. We use $M_S^0$ obtained for a bulk single crystal[7] (417 emu cm$^{-3}$ at 3 K), with which we can then derive values of $P$ from our measured $u$, which results in $P > 1$ at all temperatures (see Fig. S3). Therefore, we introduce a non-adiabatic spin-transfer torque term, $\beta$, to ensure a realistic $P$ value[29–32]. The DW mobility is then given by $u = \beta u_B P/\alpha e M_S$, where $\alpha$ is the Gilbert damping parameter[29–32], which allows $P < 1$ when $\beta > \alpha$.

Experimentally, via superconducting point contact measurements, $P$ can be determined from the point contact conductance, $G(V)$, versus applied voltage, $V$, according to the equation $2(1 - P) = \frac{G(V)}{G_n}$ ($eV \to 0$), where $G_n$ is the conductance of normal states[35]. In order to perform such an experiment, we fabricated an all 2D vdW heterostructure composed of a point contact formed from a superconducting 2H-NbSe$_2$ flake on top of a FGaT flake, as shown in Fig. 2a. From these measurements, we find a value of $P$ ~ 0.415 using the conductance values at V→0 according to the above equation (see Fig. 2b). A numerical fitting based on the modified Blonder−Tinkham−Klapwijk

(BTK) model over the entire voltage range gives similar $P$ values, as shown in Supplementary Note 2 and Fig. S4. We note that these values are comparable to those found in conventional ferromagnets such as Co and Fe[35]. From the magnitude of $P$, the ratio $\beta/\alpha$ as a function of temperature can be deduced from the measured DW mobility (see Fig. 2c). We note that $P$ will decrease with increasing temperature, especially when approaching $T_C$, thus giving a lower bound for $\beta/\alpha$. Nevertheless, the ratio $\beta/\alpha$ increases with decreasing temperature and shows a value over 1 at all temperatures, indicating that the STT driven DW motion in FGaT corresponds to a spin angular momentum transfer from the conduction electrons to the local moments that exceeds 1[29,30].

We now compare the CIDWM in FGaT and its sister material Fe$_3$GeTe$_2$ at the same temperature of 20 K (see Fig. 2d). FGaT shows much more efficient CIDWM than Fe$_3$GeTe$_2$, with a one-order of magnitude larger $u$ and a 6-times smaller $J_{th}$ (see Table 2). The large $\beta/\alpha$ ratio in FGaT as compared to that in Fe$_3$GeTe$_2$ ($\beta/\alpha \sim 0$, Ref. 26.) plays a decisive role. We note that $J_{th}$ has extrinsic and intrinsic origins. With respect to extrinsic mechanisms, both Fe$_3$GeTe$_2$ and Fe$_3$GaTe$_2$ exhibit significantly smaller values of $J_{th}$ as compared to the lowest reported values for conventional ferromagnetic multilayered heterostructures[33]. We attribute these low values to the atomic level smoothness of the van der Waals magnets. With respect to intrinsic mechanisms, when the non-adiabatic term $\beta$ is small, there exists an

intrinsic threshold current density[31,32]. The considerably smaller $\beta$ in Fe$_3$GeTe$_2$ as compared to FGaT will thus give rise to a larger intrinsic pinning.

In order to obtain values of $\beta$ and $\alpha$ the CIDWM in the presence of an out-of-plane (OOP) magnetic field is measured. The velocity of the DWM in the presence of both current and magnetic field can be expressed as the sum of the velocity induced by the current and the magnetic field as follows, $v = v_J + v_H$, where $v_J = u \times J$ and $v_H = \Delta\gamma H/\alpha$. Here, $\Delta$ is the DW width and $\gamma$ is the gyromagnetic ratio[30,36]. An exemplary CIDWM velocity as a function of the applied OOP field at 290 K is shown in Fig. 3a. The field-induced DW mobility, $\Delta\gamma/\alpha$, deduced from the OOP field dependence is plotted as a function of temperature in Fig. 3b. A high STT efficiency which increases with decreasing temperature is also confirmed by comparing the current-induced and field-induced DW mobility (see Fig. S7). The DW width at each temperature can be calculated using $\Delta = \sqrt{A/K_u^{eff}}$, where $A$ is the exchange stiffness and $K_u^{eff}$ is the uniaxial anisotropy energy[33]. Through magnetic field dependent anomalous Hall transport measurements, as shown in Fig. S5 and S6, $K_u^{eff}$ can be determined from $K_u^{eff} = \frac{1}{2}H_K^{eff} \times M_S$ (Fig. 3c). By using a value of the exchange stiffness from earlier studies[37], $A = 1.33\,\text{pJm}^{-1}$, the calculated DW width is plotted as a function of temperature in Fig. 3c. The Gibert damping parameter, $\alpha$, is then calculated from the equation $\alpha = \Delta\gamma H/v_H$ (Fig. 3d). The resulting dependence of $\alpha$ and $\beta$ on temperature is summarized in Fig. 3d. The large $\beta$ values that we find in FGaT may have the following origins: the non-adiabatic term is predicted to be larger in materials with a smaller DW width[31], as in FGaT: the DW width is calculated to be below 2 nm due to its high PMA; another possible origin is the high resistivity of FGaT, which is over 500 $\mu\Omega$ cm at all temperatures from temperature dependent resistivity measurements (see Fig. S5d). Interestingly, a Kondo-like behavior at low temperatures is observed, indicating strong electron-spin

**Table 2 | Comparison of CIDWM in Fe$_3$GeTe$_2$ and FGaT at 20 K**

|  | Fe$_3$GeTe$_2$ (Ref. 26) | FGaT (This work) |
|---|---|---|
| $T$ (K) | 20 | 20 |
| $u$ ($10^{-11}$ m$^3$ A$^{-1}$ s$^{-1}$) | 3.22 | 33 |
| $J_{th}$ (MA cm$^{-2}$) | 17.9 | 3 |

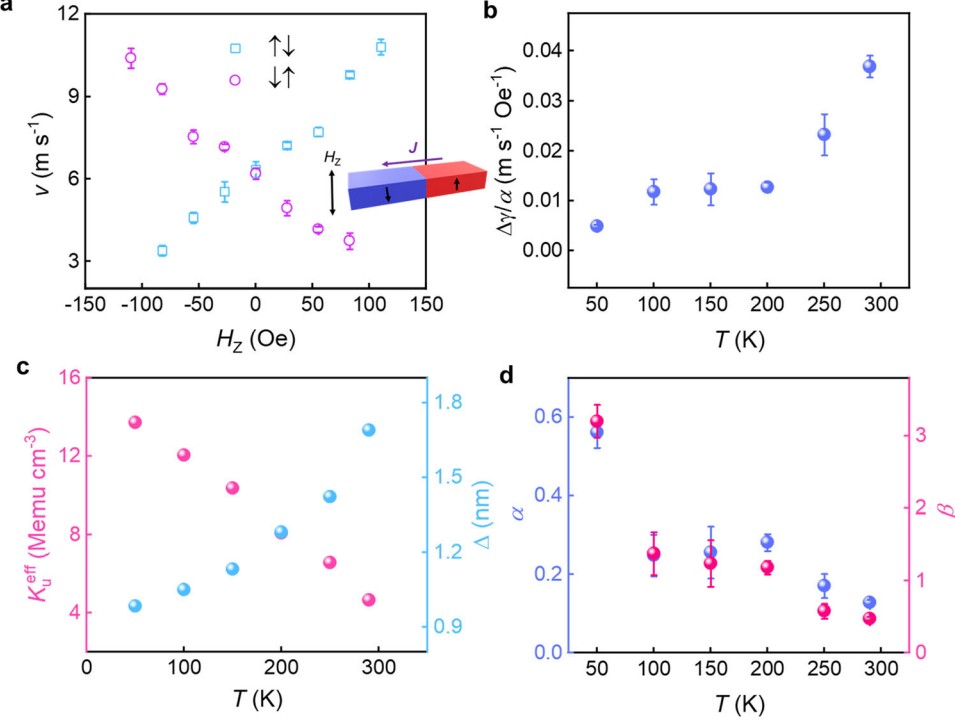

**Fig. 3 | Determination of non-adiabatic STT term. a** CIDWM for ↑↓ and ↓↑ DWs plotted as a function of exterior OOP magnetic field, $H_Z$. The DW velocities are measured at an injected current density of −3.5 MA cm$^{-2}$ at 290 K. **b** Fitted slope of DW velocity versus OOP field with the form of $\Delta\gamma/\alpha$ as a function of temperature. **c** Magnetic uniaxial anisotropy energy, $K_u^{eff}$, and calculated domain wall width plotted as a function of temperature. **d** Calculated Gibert damping parameter $\alpha$ and non-adiabatic term $\beta$ versus temperature. All error bars correspond to 1 SD.

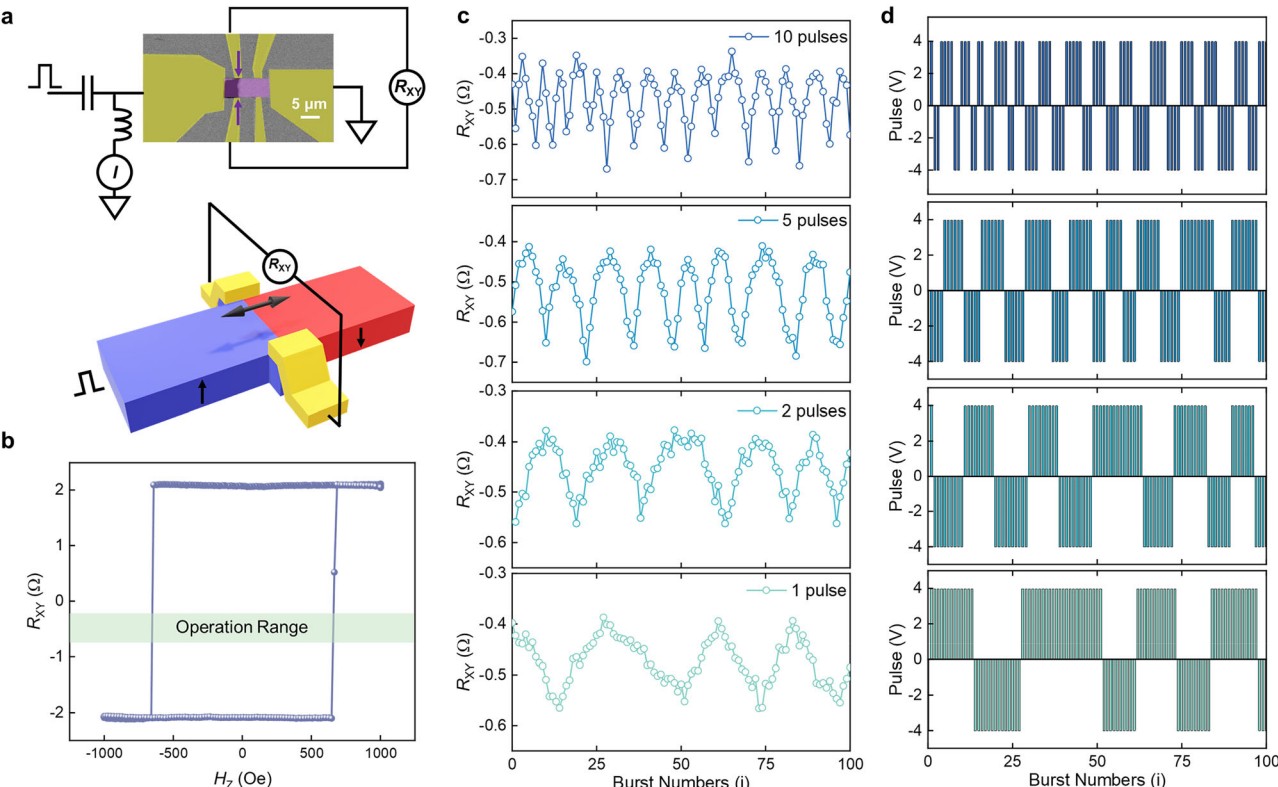

**Fig. 4 | Racetrack memristor fabricated from the van der Waals magnet Fe₃GaTe₂.** **a** Kerr microscopy image (top) and schematic of the measurement set-up (bottom) of a Racetrack memristor device, in which a DW is generated and placed within the Hall bar region. Two Hall bars are shown but only the leftmost Hall bar is used in these experiments. **b** Anomalous Hall resistance $R_{XY}$ of the Racetrack memristor device plotted as a function of exterior OOP field $H_Z$. The light green shaded region shows the $R_{XY}$ regime where DW operations defined by bursts of pulse voltages are carried out in zero magnetic field. **c** Voltage-pulse-controlled states of $R_{XY}$ by bursts of current pulses in zero magnetic field. Results for bursts with varying numbers of voltage pulses are shown (top to bottom correspond to bursts with 10, 5, 2 and 1 pulses, respectively). **d** Schematic representation of the sequences of bursts used to give rise to the results shown in (**c**).

scattering in FGaT[20,38,39]. This should give rise to a very short spin-flip length and, thus, a high $\beta$ value. We also note reports of sizable spin-orbit coupling in FGaT[40], which may also be responsible for the high $\beta$ value.

We find the characteristics of FGaT from the above experiments attractive for practical spintronic devices. Thus, we now demonstrate how the highly efficient STT-driven DWM in FGaT can be used for a vdW spintronic memristive device - something that, to the best of our knowledge, has not been previously reported in vdW materials. In order to demonstrate memristive characteristics, Hall bar detectors, which can have a spatial resolution of better than a few tens of nanometers[24], are integrated into a FGaT racetrack (see in Fig. 4a). First, a single DW is injected into the Hall bar (Fig. 4a) by applying nanosecond voltage pulses in the presence of an external OOP field (see "Methods" for more details). Once the DW is injected into the Hall bar region, the external OOP field is removed so that the following operations are conducted in zero magnetic field. The injected DW is then controlled by ns-long pulses and the motion of the DW along the racetrack induces an evolution of the anomalous Hall resistance $R_{XY}$. We note that, as shown in Fig. 4b, the device displays an $R_{XY} \sim \pm 2\,\Omega$, and thus a Hall resistance signal resolution ~2 Ω/μm, taking into account the Hall bar size of 2 μm. In Fig. 4c, we show a continuous decrease/increase of $R_{XY}$ due to the reliable backward/forward motion of a single domain wall (↑↓ DW) inside the Hall bar detector. We inject bursts of variable numbers of from 1 to 10 5 ns-long voltage pulses (± 4 V corresponding to $J \sim \pm 3$ MA cm⁻²). The interval between the individual pulses is 100 μsec and that between the bursts is 200 msec. The injected bursts for the DWM are shown in Fig. 4d. Note that the operating current density is

more than 1 order of magnitude lower than conventional magnetic thin film-based devices (e.g., $J_{th} > 50$ MA cm⁻² in Pt/Co/Ni/Co based racetrack[24]). It is clear that the number of $R_{XY}$ values, i.e., the number of data bits, can be controlled by the number of injected voltage pulses in one burst. As shown in Fig. 4c, for bursts with fewer numbers of pulses, a larger number of intermediate $R_{XY}$ values can be obtained: when using a single pulse in one burst, the number of intermediate $R_{XY}$ values can be more than 20, corresponding to the possibility of storing more than 4 data bits in one racetrack device. Considering that the operational range of $R_{XY}$ used here is only ~1/10 of the total $R_{XY}$ (shaded region in Fig. 4b), this indicates that the position of the DW within the 2 μm wide Hall bar region is controlled to within ~200 nm. Thus, this shows that the position of the DW can be controlled to within a very high spatial resolution of ~10 nm when a single pulse is used (Figs. 4c, "1-pulse" measurement). In other devices, by thinning down the thickness of FGaT, the $R_{XY}$ signal can be increased (Fig. S8) and the continuous injection of pulses can drive a single DW across multiple Hall bars (Fig. S9). We have also illustrated the reliability of our Racetrack memristor device under more than 150 bursts with back-and-forth pulses, during which no reset or regeneration of the DW is needed (Fig. S10). Such a precise spatial control, via the electrical manipulation of a single DW, shows that multi-bit racetrack memristor devices based on FGaT of great potential for advanced memory and logic applications. For practical applications, the electrical injection of domain walls without any external field will be important. This can be accomplished by, for example, field-free spin-orbit torque local switching of the magnetization in the racetrack using vdW Weyl semimetals[41,42].

## Discussion

In this work, we have investigated CIDWM in a room temperature vdW magnet FGaT. A highly efficient low-threshold and high-mobility STT-driven DWM has been observed from 50 K to room temperature. A record-high current-induced DW velocity for a vdW magnet was demonstrated. The atomic level smoothness of the vdW structure is shown to substantially reduce extrinsic pinning of a DW, while a non-adiabatic spin torque allows for DW velocities that exceed the spin momentum transfer rate from the spin-polarized conduction electrons. We further demonstrate that the CIDWM in FGaT racetrack can be utilized for an electrically readable memristive device. We successfully operated such a memristor device, showing more than 4 data bits, via the reliable consecutive backward/forward motion of DWs at low operating current densities, as low as $\pm$ 3 MA cm$^{-2}$. Our work shows functional 2D vdW spintronic devices that have significant potential both for room temperature and for cryogenic temperature applications.

## Methods

### Device fabrication

FGaT nanoflakes were mechanically exfoliated from a bulk crystal using Scotch tape onto Si/SiO$_2$ substrates. The vdW heterostructures were assembled using a polymer-based dry transfer method in a N$_2$ filled glovebox with < 3 ppm O$_2$ and < 1 ppm H$_2$O. Ti(5 nm)/Au(55 nm) electrodes were fabricated by electron-beam lithographic (EBL) patterning of resist on the flakes followed by magnetron sputtering and a standard lift-off process. The racetrack and Hall bar devices were then defined using EBL resist patterning followed by Ar ion etching.

### CIDWM measurements using Magneto optical Kerr microscopy

A CryoVac cryostat with a vacuum environment of ~5 × 10$^{-6}$ mbar was used to cool the Racetrack devices. Magnetic optical Kerr microscopy was employed to image the perpendicular magnetization of the device. The DWs were created in the Racetrack devices with the aid of an external OOP magnetic field. A pulse generator PSPL10300B was used to generate ns-long pulses to move the DWs.

### Generation of domain walls and transport measurements of a Racetrack memristor device

An exterior OOP magnetic field of–2000 Oe was initially applied to the Racetrack memristor device to magnetize it into a single domain state. Then a reduced OOP field of + 100 Oe was applied, together with the application of a train of 20 current pulses with 5 ns length and magnitude slightly smaller than the threshold current density. This process created and trapped a domain wall (DW) in the Hall bar region. Electrical transport measurements were carried out in a Lakeshore probe station. In the probe station, Be-Cu RF tips (DC - 40 GHz) were used for injecting ns-long voltage pulses into the Racetrack memristor devices. A bias-tee was used for detecting the Hall signals while the voltage pulses are injected. Keithley 6221 & 2182a instruments were used for continuous DC measurements while a pulse generator PSPL10300B was used for injecting voltage pulses. The DC measurements were conducted with a current of 1 µA which is small enough to avoid influencing the domain wall motion.

## Data availability

The data supporting this study and its findings are available within the article and Supplementary Information. Data from this study are available from the corresponding author on request.

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

## Acknowledgements

S.S.P.P. acknowledges financing from the Deutsche Forschungsgemeinschaft (DFG, German Research Foundation)–project no. 443406107, Priority Program (SPP) 2244. This study is also supported by the Samsung Electronics R&D program "Material and Device Research on Racetrack Memory".

## Author contributions

Y.G. and S.S.P.P. conceived the idea and designed the experiments. Y.W. and Y.Z. prepared the samples and fabricated the devices. Y.G., Y.Z. and W.Z. conducted the Kerr microscopy measurements. Y.W., Y.G. and J.-C.J. conducted the transport measurements. K.X. conducted the RMCD measurements. Y.G., J.-C.J. and S.S.P.P. wrote the manuscript with input from all the authors.

## Funding

## Competing interests

The authors declare no competing interests.
