## [Transparent Peer Review file · Nature Communications]

Highly efficient current-induced domain wall motion in a room temperature van der Waals magnet

Corresponding Author: Professor Stuart Parkin

Version 0:

Reviewer comments:

Reviewer #1

(Remarks to the Author)

Recommendation: Minor revision

In this work, the authors study the highly-efficient current-induced domain wall motion in a room-temperature vdW ferromagnet, Fe₃GaTe₂. They demonstrate that this behavior is driven by the spin-transfer torque (STT) mechanism. They also demonstrate the presence of a large non-adiabatic STT. Importantly, they present an electrically readable FGaT-based memory racetrack device with the good performance. I think this work is interesting, with some solid experimental results and discussions. However, there are still some little problems that should be fully addressed. My comments are as follows:

(1) In page 4, the authors claim that the low J_{th} is attributed to the atomic level smoothness of the FGaT layers that limits the spatial variation of magnetic properties. However, its sister material, Fe₃GeTe₂, is also a vdW material with atomic level smoothness. Why the J_{th} of FGaT is several times smaller than that of Fe₃GeTe₂?

(2) In Table 1, the authors compare the domain wall mobilities for some typical ferromagnetic systems with different mechanisms. Among them, the FGaT shows highest domain wall mobilities. However, may I ask if the observed domain wall mobilities in FGaT is the fastest one compared with all known ferromagnetic systems with all different mechanisms? If not, I think they should present at least one system with higher domain wall mobility than that of FGaT for avoiding the misleading. Moreover, they only contain one vdW ferromagnetic material, namely FGaT. For better comparison, I think they should include some other vdW ferromagnets in this table. And the test temperature should also be included in this table.

(3) In Fig. 4, they present the FGaT racetrack which is very interesting. At what temperature did this result occur?

Reviewer #2

(Remarks to the Author)

This manuscript reports on highly efficient current-induced domain wall motion at a room temperature in a 2D van der Waals magnet Fe₃GaTe₂ (FGaT) via spin-transfer torque (STT). Through superconducting point contact measurements, the authors have evaluated the spin polarization of the conduction electrons, concluding that the STT delivers the spin angular momentum carried by the conduction electrons. Utilizing this unique mechanism, electrically readable FGaT racetrack devices with more than 4 data bits have been successfully demonstrated. The paper exhibits strong scientific rigor and practical relevance, especially for the reports of the highest domain wall velocity yet reported in any vdW magnets, which could offer fresh applications into spintronic memory devices. Therefore, I recommend it for publication in Nature communications. However, before acceptance for publication, a minor revision is necessary to resolve some issues, listed as follows:

1. The authors conclude that STT transfers over 100% of the spin angular momentum carried by conduction electrons, implying near-perfect spin polarization in FGaT materials. To strengthen this claim, a detailed discussion of the physical origins underlying FGaT's exceptionally high spin polarization would be beneficial.
2. Given the nanoscale dimensions of the racetrack device and FGaT's high resistivity, Joule heating effects may play a non-negligible role. Could the authors estimate the operating temperature of the device during current-induced domain wall motion measurements? Furthermore, does Joule heating act as a facilitating or hindering factor for STT-driven domain wall

motion in 2D van der Waals magnets?

3. While the authors demonstrate electrically readable FGaT racetrack devices capable of storing >4 data bits, the reliance on external magnetic fields for initial domain wall injection remains a critical limitation for practical applications. To enhance technological relevance, could the authors propose a strategy for achieving fully electrical (field-free) domain wall manipulation in 2D van der Waals magnets?

4. As described, distinct Rxy states are achieved by modulating both pulse width and pulse count within individual trains. Domain wall displacement exhibits deterministic bidirectional motion (left/right), correlating with Rxy transitions between maximum and minimum values. Notably, while the total pulse count remains constant, pulse width and per-train pulse quantity serve as variable parameters: wider pulses reduce per-train pulse numbers, necessitating additional pulse sequences to complete Rxy cycles. However, clarification is needed regarding the implementation of 4-bit recording (16 discrete states). Since Rxy variations follow strictly monotonic cycles (cycling between minima and maxima), and only 20 monotonic Rxy cycles are reported, the correspondence between these cycles and the 16 required non-overlapping states warrants further elaboration.

Reviewer #3

(Remarks to the Author)

This manuscript reports a highly efficient current-induced domain wall (DW) motion in FGaT racetrack devices driven by spin-transfer torque (STT). The authors demonstrate an exceptionally low threshold current density (J_{th}) for DW displacement, attributed to the structural perfection of van der Waals (vdW) magnets. Additionally, they present an electrically readable FGaT memristive racetrack device capable of encoding more than four data bits, enabled by integrating Hall bar detectors and achieving domain wall positioning with ~10 nm precision. The results are both novel and intriguing, yet the following issues should be addressed before the manuscript can be considered for publication in Nature Communications:

1. The authors describe a protocol involving a reduced out-of-plane (OOP) magnetic field combined with 20 sub-threshold current pulses (5 ns duration) to nucleate a DW. Why does this method reproducibly generate a single DW rather than multiple or randomly distributed ones? What type of DW (e.g., Bloch, Néel) is created under these conditions?

2. The low J_{th} and its weak temperature dependence are attributed to atomic-level smoothness in FGaT layers, which suppresses spatial variation in magnetic properties. However, Fe_3GeTe_2 — a structurally similar vdW magnet— also exhibits atomic flatness but does not display comparable transport characteristics. The manuscript should clarify the underlying factors responsible for the superior STT efficiency in FGaT, beyond structural perfection.

3. The spin polarization (P) of FGaT is derived from a point-contact measurement using a simplified expression $2(1-P) = (G(V))/G_n$ ($eV \rightarrow 0$). This approximation holds only under specific idealized conditions and may introduce significant error. The authors are strongly encouraged to apply a full numerical fitting based on the modified Blonder–Tinkham–Klapwijk (BTK) model (Ref. 35) across the entire voltage range to obtain a more reliable estimate of P.

4. A large non-adiabaticity parameter β is ascribed to a narrow DW width (due to high PMA), high resistivity, and short spin-flip length from strong electron-spin scattering. Should the role of spin-orbit coupling (SOC) in FGaT also be considered? The authors should discuss whether SOC may contribute significantly to the observed β .

5. Although STT-induced DW motion is extensively characterized, data on magnetic-field-driven DW motion is lacking. A comparison of threshold values (H_x vs. J_{th}) and corresponding velocities would help discern whether the low J_{th} is specifically related to STT efficiency or reflects intrinsic DW energetics in FGaT.

6. Accurate DW positioning is critical for multi-bit operation. How do the authors ensure DWs are not pinned at Hall bar edges? Furthermore, device endurance under repeated cycling should be evaluated to assess reliability for potential memory applications.

Version 1:

Reviewer comments:

Reviewer #1

(Remarks to the Author)

The authors have addressed my questions and the paper can be accepted in this version.

Reviewer #2

(Remarks to the Author)

Reviewer #3

(Remarks to the Author)

The authors have made significant improvement to the manuscript in response to the reviewers' comments. However, I still have the following concerns regarding the reply to Question 5:

What extrapolation method did the authors use to convert the coercive field H_C obtained from RMCD into the domain wall threshold field H_{th} ? Please provide the associated error estimate and the quantitative relationship (proportionality) between the two quantities.

I recommend that the authors determine the temperature dependence of the domain wall mobility by extracting the velocity threshold curves ($v-H_{th}$ and $v-J_{th}$) for magnetic field driven and current driven motion. A direct comparison of the mobilities would allow a more rigorous assessment of the role of STT efficiency. I would recommend publication of the paper after these concerns are addressed.

Version 2:

Reviewer comments:

Reviewer #3

(Remarks to the Author)

The authors have addressed all my technical concerns. I would recommend publication of this manuscript in its present form.

Reviewer #1 (Remarks to the Author):

Recommendation: Minor revision

In this work, the authors study the highly-efficient current-induced domain wall motion in a room-temperature vdW ferromagnet, Fe₃GaTe₂. They demonstrate that this behavior is driven by the spin-transfer torque (STT) mechanism. They also demonstrate the presence of a large non-adiabatic STT. Importantly, they present an electrically readable FGaT-based memory racetrack device with the good performance. I think this work is interesting, with some solid experimental results and discussions.

We thank the reviewer for the interest in our work and for recommending publication in Nature Communications with minor revisions.

However, there are still some little problems that should be fully addressed. My comments are as follows:

(1) In page 4, the authors claim that the low J_{th} is attributed to the atomic level smoothness of the FGaT layers that limits the spatial variation of magnetic properties. However, its sister material, Fe₃GeTe₂, is also a vdW material with atomic level smoothness. Why the J_{th} of FGaT is several times smaller than that of Fe₃GeTe₂?

We thank the reviewer for these comments. This is a very interesting question. The threshold current density J_{th} of current-induced domain wall motion in racetrack devices has both extrinsic and intrinsic origins. The atomic level smoothness of van der Waals magnets is expected to decrease J_{th} with respect to extrinsic mechanisms. This should apply to both Fe₃GeTe₂ and Fe₃GaTe₂. With respect to intrinsic mechanisms when the non-adiabatic term β is small, there exists an intrinsic threshold current density (see Ref. 29 and 30). In our recent work (Ref. 26), we have observed a much less efficient current-induced domain wall motion in Fe₃GeTe₂ compared to that observed in FGaT, which we attribute to a much smaller β in Fe₃GeTe₂. Thus, an intrinsic pinning induced by the smaller β is consistent with a larger J_{th} in Fe₃GeTe₂ as compared to FGaT. We have added the above discussion on this very interesting point to the main text of our revised paper (see line 23, page 5, Fig. 2d and new Table 2).

(2) In Table 1, the authors compare the domain wall mobilities for some typical ferromagnetic systems with different mechanisms. Among them, the FGaT shows highest domain wall mobilities. However, may I ask if the observed domain wall mobilities in FGaT is the fastest one compared with all known ferromagnetic systems with all different mechanisms? If not, I think they should present at least one system with higher domain wall mobility than that of FGaT for avoiding the misleading. Moreover, they only contain one vdW ferromagnetic material, namely FGaT. For better comparison, I think they should include some other vdW ferromagnets in this table. And the test temperature should also be included in this table.

We thank the reviewer for these comments. Indeed, FGaT does not display the highest reported current-induced domain wall mobility at room temperature: for example, as reported in Ref. 33, in the synthetic antiferromagnet (SAF) system, with interfacial engineering (i.e. using very thin dusting layers (DLs), the domain wall mobility can as high as $55 \times 10^{-11} \text{ A}^{-1} \text{ m}^3 \text{ s}^{-1}$ at RT (room temperature) which is slightly higher than that which we report. Until now, FGaT is the only van der Waals magnet to have been shown to exhibit very efficient current-induced domain wall motion at room temperature.

In the other well-known van der Waals ferromagnet, Fe_3GeTe_2 , whose Curie temperature is well below room temperature, the domain wall mobility (at 20 K) is one order of magnitude smaller than that in FGaT at the same temperature (i.e. $\sim 3.22 \times 10^{-11} \text{ A}^{-1} \text{ m}^3 \text{ s}^{-1}$ at 20 K as reported in Ref. 26 for Fe_3GeTe_2 and $\sim 33 \times 10^{-11} \text{ A}^{-1} \text{ m}^3 \text{ s}^{-1}$ at 20 K for FGaT as shown in Figure. R1). Since Table 1 compares the DW mobilities of different systems at room temperature, we did not include Fe_3GeTe_2 . We have included Fig. R1 and added a paragraph to the main text to point out the much less efficient CIDWM in Fe_3GeTe_2 (see line 23, page 5, Fig. 2d and new Table 2). We are also very happy to follow the reviewer's suggestion and include the measurement temperatures in the table. Below is the modified Table 1 which can also be found in our revised manuscript.

Figure. R1 Domain wall velocity v versus injected current density J for Fe_3GeTe_2 (orange circles) and Fe_3GaTe_2 (purple circles) at 20 K.

	Permalloy Ref. 26	Co/Ni Ref. 27	Pt/Co Ref. 30	Pt/SAF Ref. 30	Pt/DL/SAF Ref. 30	FGaT This Work
Measurement temperature (K)	290	290	290	290	290	290
DW mobility ($10^{-11} \text{ A}^{-1} \text{ m}^3 \text{ s}^{-1}$)	7.33	5.13	12	26	55	37
Mechanism	Spin transfer torque	Spin transfer torque	Spin-orbit torque	Exchange-coupling torque	Exchange-coupling torque	Spin transfer torque

Table R1. Modified Table 1 in the main text.

(3) In Fig. 4, they present the FGaT racetrack which is very interesting. At what temperature did this result occur?

We thank the reviewer for the comment. We measured the FGaT racetrack at the temperature of 290 K – which makes the material promising for practical spintronic applications in the near future.

Reviewer #2 (Remarks to the Author):

This manuscript reports on highly efficient current-induced domain wall motion at a room temperature in a 2D van der Waals magnet Fe_3GaTe_2 (FGaT) via spin-transfer torque (STT). Through superconducting point contact measurements, the authors have evaluated the spin polarization of the conduction electrons, concluding that the STT delivers the spin angular momentum carried by the conduction electrons. Utilizing this unique mechanism, electrically readable FGaT racetrack devices with more than 4 data bits have been successfully demonstrated. The paper exhibits strong scientific rigor and practical relevance, especially for the reports of the highest domain wall velocity yet reported in any vdW magnets, which could offer fresh applications into spintronic memory devices. Therefore, I recommend it for publication in Nature communications.

We thank the reviewer for positive evaluation of our manuscript as “exhibits strong scientific rigor and practical relevance”, and for recommending it to publication in Nature Communications.

However, before acceptance for publication, a minor revision is necessary to resolve some issues, listed as follows:

1. The authors conclude that STT transfers over 100% of the spin angular momentum carried by conduction electrons, implying near-perfect spin polarization in FGaT materials. To strengthen this claim, a detailed discussion of the physical origins underlying FGaT’s exceptionally high spin polarization would be beneficial.

We thank the reviewer for this comment. First of all, we would like to clarify that the spin polarization in FGaT is $\sim 41.5\%$, as discussed in our original manuscript, according to our superconducting point contact measurements (Fig. 2b), which is comparable to those observed in conventional ferromagnets such as Co and Fe. In current-induced domain wall motion, the presence of non-adiabatic spin-transfer torques can lead to domain wall velocities that exceed the rate of spin angular momentum transfer, which means that the STT transfers over 100% of the spin angular momentum carried by the conduction electrons (which is $\sim 41.5\%$) to the DWs. We believe that a significant non-adiabatic torque originates from the narrow domain wall width (Fig. 3c), high resistivity (Fig. S5c) and short spin-flip length from strong electron-spin scattering in the FGaT (supplementary Note. 3), in accordance with Refs 29 and 30.

2. Given the nanoscale dimensions of the racetrack device and FGaT's high resistivity, Joule heating effects may play a non-negligible role. Could the authors estimate the operating temperature of the device during current-induced domain wall motion measurements? Furthermore, does Joule heating act as a facilitating or hindering factor for STT-driven domain wall motion in 2D van der Waals magnets?

We thank the reviewer for pointing this out. The temperature of the sample will increase during the current pulses due to Joule heating. Experimentally, it is difficult for us to measure the resistance of our racetrack device during the application of nanosecond long pulses especially since we are using a two-contact geometry for these studies. To address this question, we have made new devices (4×4 microns²) with 4 contacts. Using longer millisecond long current pulses we can then measure the resistance of the device while applying the current, R_{xx}^{pulsed} . Then, by comparing the measured DC resistance, R_{xx}^{DC} , versus temperature T of the device using small DC currents with the R_{xx}^{pulsed} - J curve, as shown in Figs. R2a and R2b, we estimate the temperature increase in our sample to be ~ 10 K for current pulses with lengths of 10 ms and a current density of 3 MA cm^{-2} at the measurement temperature of 290 K. As shown in our previous work (Bläsing et al. *Nat. Commun.* **9**, 4984 (2018)), the sample temperature increases with increasing the applied pulse length. Since we observe ~ 10 K rise with 10 ms pulse, we believe that the sample temperature rise upon an application of 5 ns pulse to be less than 10 K at a current density of 3 MA cm^{-2} . However, since the contact regions in our Racetrack device are etched before deposition of the Ti/Au electrical contacts, these regions have a thinner thickness compared to the body of the device and thus the current density is locally higher so that these regions will get hotter and serve as nucleation sites thereby limiting the highest current density we could apply.

Joule heating leads to a decreased magnetization that can make DW motion easier (smaller threshold current density) but it also limits the maximum speed that a DW can reach as Joule heating leads to a reduced spin polarization.

Figure R2. Longitudinal resistance of a FGaT device versus (a) temperature measured with DC current and (b) current density measured with 10 ms long current pulses.

3. While the authors demonstrate electrically readable FGaT racetrack devices capable of storing >4 data bits, the reliance on external magnetic fields for initial domain wall injection remains a critical limitation for practical applications. To enhance technological relevance, could the authors propose a strategy for achieving fully electrical (field-free) domain wall manipulation in 2D van der Waals magnets?

We thank the reviewer for this comment. Recently, there have been several reports on the field-free switching of magnetization in FGaT utilizing out-of-plane spin-orbit torques generated from a van der Waals Weyl semimetal, such as WTe₂ (Kajale et al. *Sci. Adv.* **10**.11: eadk8669 (2024)) and TaIrTe₄ (Zhang et al. *Adv. Mater.* **36**.41: 2406464 (2024)). Incorporating these materials could be an effective strategy for achieving fully electrical manipulation of domain walls in 2D vdW magnets. We have added a discussion about a field-free switching strategy in our revised manuscript (line 29, page 7). We have also added the above-mentioned references (Ref. 39 and 40).

4. As described, distinct R_{xy} states are achieved by modulating both pulse width and pulse count within individual trains. Domain wall displacement exhibits deterministic bidirectional motion (left/right), correlating with R_{xy} transitions between maximum and minimum values. Notably, while the total pulse count remains constant, pulse width and per-train pulse quantity serve as variable parameters: wider pulses reduce per-train pulse numbers, necessitating additional pulse sequences to complete R_{xy} cycles. However, clarification is needed regarding the implementation of 4-bit recording (16 discrete states). Since R_{xy} variations follow strictly monotonic cycles (cycling between minima and maxima), and only 20 monotonic R_{xy} cycles are reported, the correspondence between these cycles and the 16 required non-overlapping states warrants further elaboration.

We thank the reviewer for this comment. As pointed out by the reviewer, pin-pointing the DW at a given position is crucial for defining the states of a memristor. Here, we could achieve 20 distinct states. However, positioning a domain wall at a given position can be challenging in a simple racetrack for several reasons including, the possible momentum of the DW itself (Torrejon *et al. Nat. Commun.* **7**, 13533 (2016); Thomas *et al. Science* **330**, 1810 (2010)) as well as unavoidable perturbations in current induced domain wall motion including thermal effects, edge defects, inhomogeneous current density, etc. (e.g., Meier *et al. Phys. Rev. Lett.* **98**, 187202 (2007); Nam *et al. Appl. Phys. Lett.* **112**, 172401 (2018); Chauve *et al. Phys. Rev. B* **62**, 6241 (2000)). Furthermore, the device we investigated has a width on the micrometer scale, so that current induced domain wall tilting will reduce the accuracy of domain wall positioning. The DW tilting can be avoided altogether in synthetic antiferromagnetic racetracks or can be mitigated using very narrow racetracks while local pinning centers can help to precisely position the DW. Several recent studies have proposed different methods to provide extrinsic pinning sites (e.g., Lee *et al. Nat. Commun.* **14.1**: 7648 (2023); Durner *et al. ACS nano* **19**, 5, 5316 (2025)). The precise positioning of a DW will clearly be the focus of future studies.

In order to characterize the quality of the multi-bit operation of our devices, we can use statistical analysis. Figure R3 shows a histogram of the R_{xy} values over 8 cycles. Although the weights for each state are different, discrete states can be observed from the histogram. To confirm the correspondence between cycles, we executed further experiments with more than 150 burst cycles (each burst includes 5 current pulses). Figures R4a and R4b show the time dependent R_{xy} signals and the corresponding phase space plot, respectively. This plot confirms that the operation trajectories over many cycles are converging and reliable – confirming its stability and endurance (Ref. 24). In a perfect racetrack device – where there exist no pinning or thermal effects – a linear time (or iteration) series signal as a function of input pulses (ramp up and down for positive and negative pulses, respectively) can be observed, as shown in Fig. R4c. In such a case, the corresponding static phase space (recurrent) plot, equivalent to a burst mode of 3 pulses, displays

a perfectly overlapping trajectory (Fig. R4d). Due to the unavoidable perturbations including thermal effects, edge defects, inhomogeneous current density, etc, there exists non-linearity and a low level of chaoticity over repeated motion, evident in the form of a trajectory dispersion in the phase space plot (Fig. R4b). The above content has been included in our revised main text (line 24, page 7) and Supplementary Information (Fig. S9).

Figure R3. Histogram of the R_{xy} values in 1-pulse-burst case. The discrete states are statistically defined with labels.

Figure R4. (a) More than 150 cycles of FGaT racetrack device operation. Here, each burst includes 5 current pulses that are 5 ns-long spaced by 100 μ s, with a 200 ms burst period. (b) Static phase (re-current) plot of current induced domain wall motion in FGaT. This confirms the repeatable trajectory and endurance of the motion. (c) Simulated time (iteration)-series signal from an ideal racetrack without pinning. (d) Static phase plot corresponding to a perfectly linear current induced domain wall motion.

Reviewer #3 (Remarks to the Author):

This manuscript reports a highly efficient current-induced domain wall (DW) motion in FGaT racetrack devices driven by spin-transfer torque (STT). The authors demonstrate an exceptionally low threshold current density (J_{th}) for DW displacement, attributed to the structural perfection of van der Waals (vdW) magnets. Additionally, they present an electrically readable FGaT memristive racetrack device capable of encoding more than four data bits, enabled by integrating Hall bar detectors and achieving domain wall positioning with ~ 10 nm precision. The results are both novel and intriguing, yet the following issues should be addressed before the manuscript can be considered for publication in Nature Communications:

We thank the reviewer for considering our paper to be “both novel and intriguing” and for recommending it to publish in Nature Communications.

1. The authors describe a protocol involving a reduced out-of-plane (OOP) magnetic field combined with 20 sub-threshold current pulses (5 ns duration) to nucleate a DW. Why does this method reproducibly generate a single DW rather than multiple or randomly distributed ones? What type of DW (e.g., Bloch, Néel) is created under these conditions?

We thank the reviewer for this question. During the fabrication of our racetrack devices the magnetic property of the racetrack is locally modified during the preparation of the electrodes. Before depositing the Ti/Au layers that form the electrodes (in a lift-off process) on top of the FGaT flake we use an etching process to clean the surface of the flake to ensure a good electrical contact. This means that the thickness of the FGaT beneath the gold electrodes is slightly thinner than that within the body of the device itself. This modifies the switching field so that this portion of the racetrack can be more easily switched, thereby naturally allowing for a DW to be nucleated. Upon applying the field together with current pulses below the threshold, the magnetization of the thinner FGaT region switches first due to its reduced switching field, leading to the nucleation of a single domain wall. This domain wall then moves into the racetrack device by the following current pulses and gets trapped at the Hall bar site due to the local dispersion of the current density in the Hall bar. Thus, this current-assisted field-induced domain creation protocol, followed by domain wall propagation allows for the reliable injection of a single DW.

In the FGaT system, bulk Dzyaloshinskii-Moriya interaction (DMI) is confirmed as there have been previous works reporting the existence of Néel type skyrmions in this system (see Ref. 16-18). As for its sister material Fe_3GeTe_2 , the DMI in FGaT originates from a broken inversion symmetry due to Fe vacancies in the van der Waals layers and additional Fe atoms in the van der Waals gaps (Ref. 18). Such a

DMI leads to a Néel type domain wall in FGaT, which is further confirmed from the in-plane longitudinal field H_x dependence of the current-induced domain wall velocity v . Figure R5 shows dome-like v - H_x curves that are offset to positive and negative fields for up/down and down/up domain walls. This behavior is a typical feature of Néel type domain walls and is well described using the well-known one-dimensional (1D) model (see Ref. 26 and Filippou et al. *Nat. Commun.* **9**, 4653 (2018)) with parameters obtained from our measurements (see Table R2). We have included the above content in the revised main text (see line 17, page 3) and Supplementary Information (see Supplementary Note 1, Fig. S2 and Table S1).

Figure R5. The in-plane longitudinal field dependence H_x of the domain wall velocity v in a FGaT racetrack device at 290 K. The pink and light blue open circles correspond to up/down and down/up domain wall configurations, respectively. The solid lines are the 1-D model fit. Current pulses with a current density of 3.2 MA cm^{-2} and a pulse length of 5 ns are used.

Domain wall width (nm)	Damping parameter	Non-adiabatic term	Shape anisotropy field (Oe)	Effective DMI field (Oe)	Saturation magnetization (emu cm^{-3})
1.7	0.13	0.5	1200	180	240

Table R2. Fitting parameters used in the 1D model fit in Figure R2.

2. The low J_{th} and its weak temperature dependence are attributed to atomic-level smoothness in FGaT layers, which suppresses spatial variation in magnetic properties. However, Fe_3GeTe_2 —a structurally similar vdW magnet—also exhibits atomic flatness but does not display comparable transport characteristics. The manuscript should clarify the underlying factors responsible for the superior STT efficiency in FGaT, beyond structural perfection.

We thank the reviewer for this question which is the same question from reviewer #1. We give the same response. This is a very interesting question. The threshold current density J_{th} of current-induced domain wall motion in racetrack devices has both extrinsic and intrinsic origins. The atomic level smoothness of van der Waals magnets is expected to decrease J_{th} with respect to extrinsic mechanisms. This should apply to both Fe_3GeTe_2 and Fe_3GaTe_2 . With respect to intrinsic mechanisms when the non-adiabatic term β is small, there exists an intrinsic threshold current density (see Ref. 29 and 30). In our recent work (Ref. 26), we have observed a much less efficient current-induced domain wall motion in Fe_3GeTe_2 compared to that observed in FGaT, which we attribute to a much smaller β in Fe_3GeTe_2 . Thus, an intrinsic pinning induced by the smaller β is consistent with a larger J_{th} in Fe_3GeTe_2 as compared to FGaT. We have added the above discussion on this very interesting point to the main text of our revised paper (see line 23, page 5, Fig. 2d and new Table 2).

3. The spin polarization (P) of FGaT is derived from a point-contact measurement using a simplified expression $2(1-P) = G(V)/G_n$ ($eV \rightarrow 0$). This approximation holds only under specific idealized conditions and may introduce significant error. The authors are strongly encouraged to apply a full numerical fitting based on the modified Blonder–Tinkham–Klapwijk (BTK) model (Ref. 35) across the entire voltage range to obtain a more reliable estimate of P .

We appreciate the encouragement from the reviewer and have carried out the numerical fitting based on the BTK model to our data, as shown in Fig. R6a. The fitting parameters used are shown in Table R3. The two dips at bias voltage of $\sim \pm 2$ meV are not considered for the fitting since there exists no superconducting gap that can account for the position of these dips. P value obtained from the fitting (~ 0.38) shows a similar value to that obtained in our manuscript using the approximation (~ 0.415). We have also conducted the BTK fitting using a two-gap model for the 2H-NbSe₂ as reported by former literatures (Guillamón et al. *Phys. Rev. Lett.* **101**, 166407 (2008); Noat et al. *Phys. Rev. B* **92**, 134510 (2015)), which gives a P value of ~ 0.42 (see Fig. R6b and Table R4). Thus, we believe the P value we used in our manuscript is reliable. We have included the above content in the revised main text (line 13, page 5) and Supplementary Information (Supplementary Note 2, Fig. S4 and Table S2 and S3).

Figure R6. Fitting of the normalized conductance to bias voltage curve from the superconducting point contact measurement using a one-gap BTK model (a) and a two-gap BTK model (b).

Δ (meV)	Γ (meV)	Z	P	T (K)
0.7	0.01	0.15	0.38	2

Table R3. Fitting parameters used for the one-gap BTK model.

Δ_1 (meV)	Γ_1 (meV)	Z_1	Δ_2 (meV)	Γ_2 (meV)	Z_2	P	T (K)
1.1	0.65	0	0.6	0.4	0.9	0.42	2

Table R4. Fitting parameters used for the two-gap BTK model.

4. A large non-adiabaticity parameter β is ascribed to a narrow DW width (due to high PMA), high resistivity, and short spin-flip length from strong electron-spin scattering. Should the role of spin-orbit coupling (SOC) in FGaT also be considered? The authors should discuss whether SOC may contribute significantly to the observed β .

We thank the reviewer for this insightful comment. As pointed out by the reviewer, there have been theoretical calculations of spin-orbit coupling in FGaT (Cho, Woohyun, et al. *Adv. Mater.* **36**.31: 2402040 (2024)). According to Cho, Woohyun, et al., the strong electron-spin scattering in FGaT originates from its sizable spin-orbit coupling. It will be interesting to investigate if there exist a correlation between spin-orbit coupling and the β term, and can thus provide a strategy of designing materials for more efficient STT-driven domain wall motion. We are very happy to include the following sentence in line 23, page 6: “We also note reports of sizable spin-orbit coupling in FGaT, which may also be responsible for the high β value”, and we have added Cho, Woohyun, et al. *Adv. Mater.* **36**.31: 2402040 (2024) as Ref. 38 in the main text. However, considering the lack of experimental methods to quantify the spin-orbit coupling, especially from domain wall motion measurements, a detailed discussion on whether spin-orbit coupling may contribute significantly to β term is beyond the scope of our current research.

5. Although STT-induced DW motion is extensively characterized, data on magnetic-field-driven DW motion is lacking. A comparison of threshold values (H_x vs. J_{th}) and corresponding velocities would help discern whether the low J_{th} is specifically related to STT efficiency or reflects intrinsic DW energetics in FGaT.

We thank the reviewer for this comment. Figure R7a shows the threshold field H_{th} and the threshold current density J_{th} , for domain wall motion at two different temperatures. Due to the technical limitations of our Kerr microscope, we cannot apply an out-of-plane field above 1100 Oe. This makes us unable to correctly estimate H_{th} once the temperature is reduced to 200 K. Nevertheless, it is clear that as the temperature is decreased from 290 K to 250 K, H_{th} increases by a factor of ~ 3 while the threshold current density J_{th} , on the contrary, decreases to $\sim 60\%$. We can infer the temperature dependence of H_{th} from the coercive field H_C obtained from Reflective Magnetic Circular Dichroism (RMCD) measurements (see Figure R7b). H_C increases over 8 times when the temperature decreases from 290 K to 50 K, while J_{th} only increases by 33% (See Fig. 1f in Main text). As discussed in the main text, the DW mobility induced by an external magnetic field shows an 8-fold decrease when the temperature is decreased from 290 K to 50 K (Fig. 3b), while the DW mobility induced by current remains almost unchanged (Fig. 1e). Based on these results, we believe that the low J_{th} is specifically related to the STT efficiency.

Figure R7. (a) Temperature dependence of the threshold field H_{th} and threshold current density J_{th} of the domain wall motion in FGaT racetrack device. (b) Temperature dependence of the coercive field H_C obtained from RMCD measurements.

6. Accurate DW positioning is critical for multi-bit operation. How do the authors ensure DWs are not pinned at Hall bar edges? Furthermore, device endurance under repeated cycling should be evaluated to assess reliability for potential memory applications.

We thank the reviewer for these comments. We confirm the DWs are not pinned at the Hall bar edge from the experiments shown in Fig. S8, where a single DW can travel freely from one Hall bar to the other back and forth by applying certain trains of nano-second current pulses with higher current density in a similar device. Also, as shown in the static space plot below (Figure. R8b), it exhibits a typical homogeneous DW motion instead of a pinned DW (see Ref. 24), so we believe the DW is not pinned at the Hall bar edge. Note that if there exists a pinning site within the detection window, one should observe a pinch point within the phase space plot. When the device is scaled down (e.g., sub-100 nm racetrack channel), and the length of the Hall bar becomes comparable to the width of the racetrack, the Hall bar can act as a relatively strong pinning site (Ref. 24). While this is not necessarily a disadvantage, it can be utilized as highly precise domain wall stopping point (Lee, *et al. Nat. Commun.* **14.1**: 7648 (2023); Durner, *et al. ACS nano* **19**, 5, 5316 (2025)).

Furthermore, the phase space plot shows over-lapping data (with limited level of data dispersion) over the many cycles of the operation, indicating predictable and stable motion. During our experiments, once the DW is injected in the racetrack memristor device, it can survive repeated cycles under more than 150 bursts with back-and-forth pulses (see Figure R8a). No reset or regeneration of the domain wall were carried out during the whole experiment after the domain wall was created and injected into the racetrack memristor device. In an ideal racetrack device – where there exist no pinning or thermal effects – a perfectly linear time (or iteration) series signal can be observed, as shown in Fig. R8c. In such a case, the corresponding static phase space (recurrent) plot, equivalent to a burst mode of 3 pulses, displays a perfectly overlapping trajectory (Fig. R8d). In reality, there exist unavoidable perturbations including thermal effects, edge defects, inhomogeneous current density, etc. (e.g., Meier *et al. Phys. Rev. Lett.* **98**, 187202 (2007); Nam *et al. Appl. Phys. Lett.* **112**, 172401 (2018); Chauve *et al. Phys. Rev. B* **62**, 6241 (2000)). These perturbations introduce non-linearity and a low level of chaoticity over repeated motions, evident in the form of a trajectory dispersion in the phase space plot (Fig. R8b). The above content has been included in our revised main text (line 24, page 7) and Supplementary Information (Fig. S9).

Figure R8. (a) Evolution of anomalous Hall resistance in racetrack memristors with burst numbers over 150. (b) Static phase space plot of R_{XY} corresponding to the injecting of bursts each with 5 current pulses. (c) Simulated time (iteration)-series signal from an ideal racetrack without pinning. (d) Static phase plot of a perfectly linear current induced domain wall motion.

Reviewer #3 (Remarks to the Author):

The authors have made significant improvement to the manuscript in response to the reviewers' comments. However, I still have the following concerns regarding the reply to Question 5:

What extrapolation method did the authors use to convert the coercive field H_C obtained from RMCD into the domain wall threshold field H_{th} ? Please provide the associated error estimate and the quantitative relationship (proportionality) between the two quantities.

I recommend that the authors determine the temperature dependence of the domain wall mobility by extracting the velocity threshold curves ($v-H_{th}$ and $v-J_{th}$) for magnetic field driven and current driven motion. A direct comparison of the mobilities would allow a more rigorous assessment of the role of STT efficiency. I would recommend publication of the paper after these concerns are addressed.

We thank the reviewer for recommending publication of our manuscript after we address one final matter which we are very happy to do.

As we mentioned in our previous response to question 5 from the same reviewer in our prior rebuttal, there is an experimental limitation in our Kerr microscope system in that we cannot apply an out-of-plane field over 1100 Oe. Thus, we cannot directly measure H_{th} below a certain temperature (200 K). Rather we inferred the temperature dependence of H_{th} from the temperature dependence of H_C obtained from RMCD measurements on the same device (in a different cryostat – an Opticool system). We find that H_C is typically larger than H_{th} which makes sense since the former relates to the nucleation of a DW whereas the latter refers to the motion of an already existing DW. Moreover, it has previously been reported that there is relation between these 2 quantities, for example, see Metaxas et al. *Phys. Rev. Lett.* **99**, 217208 (2007). In Table R1 we compare our measured H_{th} and H_C at two different temperatures, 250 K and 290 K.

The reviewer asks us to assess more rigorously the STT efficiency. We are happy to do this. We can calculate the STT efficiency from our measurements of domain wall mobilities induced by current and field, namely mobility- J and mobility- H , respectively (see Fig. R1a: these data are taken from Fig. 1e and 3b of the main text). The domain wall mobilities are defined as the derivative of the domain wall velocity with respect to the current or the field i.e. $\frac{dv}{dJ}$ or $\frac{dv}{dH}$.

The STT efficiency can be defined as mobility- J /mobility- H as, for example, recently discussed in Wang et al. *Nat. Electro.* **6**, 119-125 (2023). The STT efficiency for our sample is then shown in Fig. R1b.

Note that we do not need to find H_{th} in order to determine the STT efficiency for the following reason. Since the current- and field-induced domain wall velocities are additive to each other (see Koyama, T., et al. *Nat. Nanotechnol.* **7**, 635-639 (2012)), as long as the domain wall is moving, we can find the mobility- H even when the applied field is smaller than H_{th} .

In conclusion, we find a high STT efficiency from the direct comparison of the mobility- J and mobility- H . We have included the above content as Supplementary Note 4 and Fig. R1b as Fig. S7 in our revised supplementary information and added contents concerning the STT efficiency in our revised manuscript (see line 10-11, page 6).

T (K)	H_C (Oe)	H_{th} (Oe)	Proportionality
290	509 ± 3	220 ± 12	$43 \pm 2.3\%$
250	1257 ± 51	715 ± 30	$56.9 \pm 3.3\%$

Table. R1 Comparison of coercive field and threshold field of domain wall motion at 250 K and 290 K.

Figure. R1 Spin-transfer torque (STT) efficiency calculated from the comparison between the current- and field-induced domain wall mobility. (a) the current-induced domain wall mobility, namely mobility- J , and field-induced domain wall mobility, namely mobility- H as a function of temperature; (b) calculated STT efficiency as a function of temperature.